# Human health risk assessment of arsenic and potentially toxic elements exposure in bread and wheat flour in Northeast Iran

Fateme Asadi Touranlou[1,2], Sajjad Ghasemi[3], Mahdi Gholian Aval[4,5], Mohammad Hashemi[1,2]*[☉], Seyedeh Belin Tavakoly Sany [iD][5,6]*[☉]

1 Medical Toxicology Research Center, Mashhad University of Medical Sciences, Mashhad, Iran,
2 Department of Nutrition, Faculty of Medicine, Mashhad University of Medical Sciences, Mashhad, Iran,
3 Department of Food Science and Technology, Faculty of Agriculture, Ferdowsi University of Mashhad, Mashhad, Iran, 4 Department of Health Promotion and Education, School of Health Mashhad University of Medical Sciences, Mashhad, Iran, 5 Social Determinants of Health Research Center, Mashhad University of Medical Sciences, Mashhad, Iran, 6 Department of Health, Safety and Environment Management, School of Health, Mashhad University of Medical Sciences, Mashhad, Iran

☉ These authors contributed equally to this work.
* belintavakoli332@gmail.com, tavakkolisanib@mums.ac.ir (SBTS); hashemimd@mums.ac.ir (MH)

## Abstract

### Background

Bread, a staple food, is at risk of contamination from heavy metals that pose significant health hazards. This study aimed to assess the levels of heavy metals, including Cu, Zn, Fe, Al, Co, As, Hg, Cr, Ni, Pb, Cd, and V, in bread samples collected from bakeries in Mashhad, Iran, and to evaluate the associated health and carcinogenic risks.

### Methods

In this study, the urban area was divided into five regions using simple random sampling. A total of 270 samples, including flour, dough, and bread, were collected from bakeries between June and December 2020. After preparing the samples and applying an acid digestion method, the concentration of heavy metals were determined using an inductively coupled plasma optical emission spectrometer.

### Results

Levels of Al, As, Cr, and Fe exceeded WHO and FAO thresholds in all regions. The health risk assessment indicated that non-carcinogenic risks from arsenic and the mean hazard index for bread consumption exceeded 1 in both adults and children, suggesting potential health risks. Moreover, carcinogenic risk indices for both age groups surpassed the acceptable limit (CR > 1 × 10⁻⁴) across all regions, indicating potential carcinogenic effects.

**Data availability statement:** Data are available within the paper or as supplementary information.

**Funding:** This study was funded by Mashhad University of Medical Sciences, Deputy of Research, Grant No.971851.

**Competing interests:** The authors have declared that no competing interests exist

## Conclusion

The findings reveal significant health and carcinogenic risks linked to bread consumption in Mashhad. Continuous monitoring of heavy metal levels in wheat and related food products is recommended to protect public health.

## Introduction

Food contamination with arsenic and heavy metals due to factors such as bioaccumulation, persistence in the environment, longevity, and toxicological characteristics significantly impacts human health and can lead to various diseases and complications. This issue is one of the foremost concerns regarding food security and safety in human societies [1,2]. In this context, organizations such as the European Commission (EC), the World Health Organization (WHO), Food and Agriculture Organization (FAO), and various regulatory bodies in different countries strictly regulate the permissible concentration or maximum allowable limits of toxic heavy metals in food products [3].

Risk assessment, as developed by the U.S. Environmental Protection Agency (EPA), provides a framework for estimating potential health risks from environmental pollutants. It involves identifying and quantifying the likelihood of adverse health effects from specific chemical exposures, considering all relevant exposure pathways [4]. According to WHO and the International Programme on Chemical Safety, risk assessment estimates the probability of adverse health effects from current and future exposure to environmental chemical pollutants in organisms, systems, or populations [5].

Heavy metals such as arsenic occur naturally but can also originate from human activities like industrial processes, agriculture, mining, and traffic emissions [6,7]. The primary exposure pathway for humans is through consumption of contaminated foods [8,9]. Bread, a staple worldwide rich in energy, fiber, and nutrients [10–13], is particularly vulnerable to contamination from soil, irrigation, fertilizers, milling, and processing additives (e.g., water, salt) [14,15].

Iran ranks second globally in per capita bread consumption, with an intake approximately double that of European nations [16]. Global statistics indicate an average daily bread consumption of 330–410 grams per person, whereas in Iran, this figure is elevated, averaging 420 grams per person [17]. Common bread types in Mashhad include traditional Iranian varieties such as Sangak (a whole wheat sourdough baked on pebble stones), Barbari (a thick, yeast-leavened flatbread), Taftoon (a thin, soft flatbread), and Lavash (a thin, unleavened flatbread), which are produced in both traditional and semi-industrial bakeries with wheat flour [18].

Sources of contamination in flour and bread are diverse, encompassing agricultural, industrial, and post-harvest factors. During wheat cultivation, heavy metals may enter crops through contaminated soil, often influenced by natural geochemical processes or anthropogenic activities such as mining, industrial emissions, and urban runoff [19]. Irrigation water, particularly groundwater, is a major source of arsenic in

wheat, especially in regions like Iran where contaminated aquifers are used for agriculture [20]. The application of phosphate fertilizers and pesticides further introduces cadmium and lead into soils, which are taken up by wheat plants [21]. Beyond raw materials, individual ingredients used in bread production can contribute to heavy metal levels. Flour, as the primary component, is the dominant contamination pathway due to its direct link to wheat cultivation, but other ingredients such as salt, yeast, and food additives may also play a role. Salt, often sourced from natural deposits or seawater, can contain trace amounts of heavy metals like lead and cadmium, depending on its purity and origin [22]. Yeast, while typically low in heavy metals, may introduce contaminants if produced in facilities with contaminated water or substrates. Food additives, such as improvers or preservatives, can contain heavy metals as impurities, particularly if derived from mineral-based sources [23]. Post-harvest processing, including milling and baking, can further contribute through contact with metal-containing equipment or contaminated water used in dough preparation [24].

Research providing detailed insights into heavy metal levels, daily intake, and associated risks in bread is crucial for safeguarding public health and improving food safety policies. While previous studies in Iran have extensively examined heavy metals in wheat and related crops [25–27], research specifically focused on bread remains limited, particularly in metropolis as Mashhad. This study addresses a significant gap by providing the first health risk assessment of heavy metal exposure from bread consumption in Mashhad, a major city in Northeast Iran. Despite Iran's high per capita bread consumption and the vulnerabilities of local bakeries, no such investigation has been conducted in this region to date. By evaluating heavy metal concentrations, intake levels, and risk indices, this research extends beyond previous studies focused solely on wheat, offering new insights to improve food safety and public health in a uniquely challenged urban environment.

## Materials and methods

### Description of the study area

Mashhad, the capital of Khorasan Razavi Province, is the second-largest metropolitan area in Iran. It is located in the northeastern part of the country, situated between longitudes 58° and 20 minutes to 60° and 8 minutes, and latitudes 35° and 40 minutes to 36° and 3 minutes. The region shares borders with Turkmenistan and Afghanistan (see Fig 1) [6]. The city of Mashhad spans an area of 16,500 square kilometers and is bordered by the Binalood Mountains to the south, the Hezar Masjed Mountain range to the north, the Jomrod River basin to the southeast, and the Atrak River basin to the northwest [28,29].

Mashhad has a population of around 3.004 million, with 97.5% identifying as Persian and 2% (approximately 54,000) consisting of Pakistani, Arab, Afghan, and Turk individuals. The area's climate is categorized as cold or dry [6]. Due to the rapid growth of industrial areas, especially leather goods, fertilizers, dyeing, metal products, textiles, and chemicals, has led to a rise in various environmental pollutants associated with agricultural practices in the region [29,30].

### Sampling, storage, and transportation

Sampling was conducted based on the scientific sampling method with the aim of covering all areas in Mashhad city. Five locations (North, South, East, West, and Central) with 90 bakeries were selected for bread sampling based on their location and use. The sample size was calculated using a factorial approach. Considering the inclusion of 90 bakeries and three types of samples from each bakery (flour, dough, and bread), the total sample size for this research is 270 samples. Samples were collected from traditional and industrial bakeries in various regions of Mashhad, Iran, between June and December 2020. All bread samples were transported under dark and cool conditions and stored at a temperature of 4 °C until laboratory analysis. During that time, official analytical methods (AOAC, 2012) were applying for sample preparation in the laboratory [31]. To reduce moisture and ensure a consistent weight, precisely 10 g of each bread sample were subjected to drying in an oven set at 100 °C. Subsequently, all dried samples were blended, sifted through a 2-mm sieve,

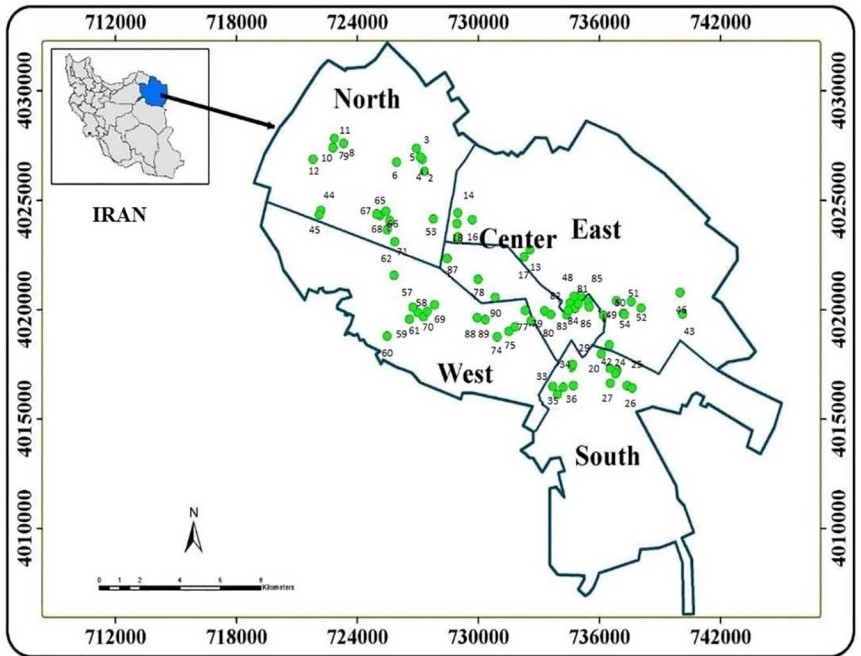

**Fig 1. Location of sampling areas in Mashhad city.**

placed in polyethylene containers that had been previously washed twice with distilled water, and stored at room temperature for further analysis [31,32].

## Chemical analysis

All standard solutions (including internal standard solutions, multi-element solutions, and standard stock solutions), reagents, and acids were obtained from Merck, Germany. The microwave digestion process (EPA 3050B method) was used to extract all dried samples [33]. Approximately 1 g of each dried sample was digested using a mixed solution of $HNO_3$ and HCl in a 1:1 ratio at a temperature of 90 °C until a clear solution was obtained. After cooling, the digested samples were filtered using filter papers (pore size: 0.22 micrometers, Whatman filters, UK) and diluted with deionized water to a final volume of 20 ml. The heavy metal concentrations (Fe, Zn, Cu, Co, Cr, V, Al, Ni, As, Hg, Pb, and Cd) in the filtered solutions were determined by inductively coupled plasma optical emission spectroscopy (ICP-OES, Spectro Arcos, model 76004555, Germany). To obtain calibration curves, both blank and standard solutions of metal ions were used [32]. Prepared standards were utilized along with a calibration blank to set the "zero" point for the ICP-OES and to confirm that there were no interferences affecting the analytical signal. In accordance with previous research and the established limit of detection (LOD), standard solutions with varying concentrations of the elements were prepared to include a range of metal ions. The correlation coefficients for the calibration curves corresponding to each element exceeded 0.99.

## Quality control and assurance

All glass bottles and containers were cleaned with diluted nitric acid ($HNO_3$) for 24 hour, then rinsed with deionized water. Subsequently, the samples were allowed to dry at room temperature before analysis. The heavy metal detection limits in the samples were estimated using a standard method. The detection limits for each element were as follows: Fe (0.06 mg/kg), Zn (0.1 mg/kg), Cu (0.1 mg/kg), Co (0.02 mg/kg), Cr (0.06 mg/kg), V (0.06 mg/kg), Al (0.18 mg/kg), Ni (0.012 mg/kg),

As (0.02 mg/kg), Hg (0.01 mg/kg), Pb (0.01 mg/kg), and Cd (0.01 mg/kg). To calibrate the ICP-OES, standard solutions of metal ions and blank were used. In order to ensure the reproducibility of the analysis, the heavy metal contents in each sample were measured in triplicate. To verify the accuracy and precision of the measurements, spiked solutions (reference solutions with known metal concentrations) and certified reference materials (CRMs), NIST 1567a (Wheat Flour), were applied. A control sample was analyzed after each batch of 10 samples to ensure an accurate analysis. The recovery percentage for all heavy metals was within an acceptable range (80.56% to 100.47%) (S1 Table), indicating the accuracy and reliability of the laboratory method. The accepted recovery range is 80%–120% [32,34]. All As and other element concentrations were reported as mg/kg on a dry weight basis. The analyses were performed at the Central Laboratory of Ferdowsi University of Mashhad.

## Human health risk assessment

According to the United States Environmental Protection Agency (EPA), increased exposure to As and heavy metals such as Fe, Zn, Cu, Co, Cr, V, Al, Ni, Hg, Pb, and Cd through bread may pose non-cancerous and cancerous health risks to humans. Over the past years, the EPA has proposed the Human Health Risk Assessment (HHRA) model evaluates the possible health risks associated with pollutants by measuring exposure and determining toxicity [35,36]. Variables used in the risk assessment equations, including ingestion rates, exposure durations, body weights, reference doses (RfD), and oral slope factors (OSF), are detailed in Table 1.

**Exposure assessment.** Chronic Daily Intake (CDI) was determined to assess the amount of arsenic and heavy metals that humans are exposed to through direct consumption, utilizing Equation (1), which is based on the EPA guidelines from 2004 and 2005 [35,36]. The estimates were made separately for children, who are considered a vulnerable group, and for adults, representing the general population.

$$CDI = (C \times IR \times EF \times ED)/BW \times AT \tag{1}$$

Where:

- CDI = Chronic Daily Intake (mg/kg/day)

- C = Concentration of the contaminant in the food item (mg/kg)

**Table 1. Health risk assessment variables for non-carcinogenic and carcinogenic risks.**

| Variables | Definition | Unit | Value | Reference |
|---|---|---|---|---|
| C | Concentration of heavy metals | mg/kg | – | – |
| IR | Ingestion Rate | kg/mg/day | Adult = 0.420 and Child = 0.210 | [41] |
| EF | Exposure Frequency | days/year | 365 | [42] |
| ED | Exposure Duration | year | Adult = 70 and Child = 6 | [42] |
| BW | Body Weight | kg | Adult = 70 and Child = 20 | [42] |
| AT | Average Time | days | Adult = 10550 Child = 2100 | [42] |
| CDI | Chronic Daily Intake | mg/kg/day | – | [42] |
| RFD | Oral Reference Dose | mg/kg/day | As = 0.003, Cd = 0.001, Al = 0.7, Co = 0.0004, Cu = 0.04, Fe = 0.7, Hg = 0.0004, v = 0.009, Zn = 0.3, Cr = 0.003, Ni = 0.2, Pb = 0.0035 | [42] |
| OSF | Oral Slope Factor | mg/kg/day | As = 1.5, Pb = 0.0085, Cd = 0.38 | [42] |
| HQ | Hazard Quotient | – | – | [42] |
| HI | Hazard Index | – | – | [42] |
| CR | Cancer Risk | – | – | [42] |

- IR = Intake Rate (kg/day) – the amount of food consumed per day

- EF = Exposure Frequency (days/year) – the number of days the food is consumed

- ED = Exposure Duration (years) – the number of years the individual is exposed

- BW = Body Weight (kg) – the weight of the individual

- AT = Averaging Time (days)

**Non-carcinogenic risk assessment.** Equations (2) and (3) were used to estimate non-carcinogenic risks using the Target Hazard Quotient (THQ) and Hazard Index (HI) [37]. The THQ represents the ratio between the reference dose (RfD) and the CDI of each element. In this study, the RfD for each element was derived from the EPA screening levels [37,38]. A population is considered to be at risk when the Hazard Quotient (HQ) is less than 1 [35,36].

$$THQ = CDI/RFD \qquad (2)$$

$$Total\ THQ\ (HI) = \Sigma THQ \qquad (3)$$

**Carcinogenic risk.** Carcinogenic risk (CR) was estimated using Equation (4) [39,40], defined as the probability of developing cancer over a lifetime due to heavy metal exposure:

$$CR = CDI \times OSF \qquad (4)$$

Where:
CR = Carcinogenic Risk, the incremental lifetime cancer probability;
OSF = Oral Slope Factor (per mg/kg/day), a measure of carcinogenic potency
The U.S. EPA acceptable range is $1 \times 10^{-6}$ to $4 \times 10^{-6}$; values > $4 \times 10^{-6}$ indicate unacceptable risk requiring intervention.

## Uncertainty and sensitivity analysis

Conventional method of risk assessment typically estimate and report risk as a single value, which fails to convey any information regarding the uncertainty of the model and its outcomes [32,43]. To gain a clearer understanding of the risk level, the EPA recommends applying the Monte Carlo simulation method. This technique utilizes mathematical statistics and probability theory to represent uncertainty through random sampling and probability distributions for each input variable. In this research, Monte Carlo simulation was applied as a probabilistic approach to minimize uncertainties. The Crystal Ball software (version 11.1.34190) facilitated the Monte Carlo modeling, conducting 10,000 iterations at a 95% confidence level. The simulated model considered the 95th percentile health risk measure, hazard index, and carcinogenic risk. Additionally, the sensitivity analysis feature of the Crystal Ball software was utilized in this study to assess the impact of each variable in the risk assessment [32,44].

Conventional methods of risk assessment usually provide a single value to represent risk, which does not adequately reflect the uncertainty associated with the model and its results [32,43]. To better understand risk levels, the EPA suggests using the Monte Carlo simulation approach. This approach applies mathematical statistics and probability theory to illustrate uncertainty by using random sampling and probability distributions for variables. In the present study, Monte Carlo simulation was applied as a probabilistic method to reduce uncertainties. The Monte Carlo modeling was conducted using Crystal Ball software (version 11.1.34190), running 10,000 iterations with 95% confidence level. The simulated model incorporated the 95th percentile hazard index measure, and carcinogenic risk. Furthermore, the sensitivity analysis

feature of the Crystal Ball software was applied in this research to evaluate the influence of each variable within the risk assessment [32,44].

## Statistical analysis

The statistical analysis was conducted using SPSS software. The Kolmogorov-Smirnov test was utilized to evaluate the normality of the results. For comparing the heavy metal concentrations in the bread samples, one-way ANOVA was used for data that followed a normal distribution, while the Kruskal-Wallis test was used for data that did not. Data description was accomplished using descriptive statistics (frequency, mean, and standard deviation) and analysis of variance (to compare variations among variables).

## Ethics approval

The study protocol was approved by the Ethics Committee of Mashhad University of Medical Sciences (IR.MUMS.MEDICAL.REC.1398.623) after obtaining the required permit for the research.

## Result

### Heavy metals concentration

The mean concentrations of heavy metals in flour, dough, and bread across five regions are summarized in S2 Table. Results showed that, among the 12 heavy metals studied, mercury (Hg) and lead (Pb) were below detection limits in all samples. The average concentrations of arsenic (As), cobalt (Co), chromium (Cr), nickel (Ni), and vanadium (V) in the flour, dough, and bread samples varied significantly between regions ($p < 0.05$). Conversely, no significant differences were observed for aluminum (Al), copper (Cu), cadmium (Cd), iron (Fe), and zinc (Zn) ($p > 0.05$). Notably, As and Ni levels were highest in the eastern, southern, and central regions. Cr concentrations peaked in the central regions, Co was most prominent in the southern region, and V showed elevated levels in both the central and western regions.

The results indicate that over 70% of the heavy metals (excluding Co) enter the bread through flour. The average heavy metal contributions from contaminated flour, dough, and the type of oven is 83%, 14.5%, and 2.5%, respectively (S1 Fig).

Based on data from S3 Table, no significant differences ($p > 0.05$) were observed in heavy metal concentrations among the different bread types available in Mashhad, including Barbari, Sanga, and Lavash.

Comparison with national and international standards (S4 Table) revealed that the concentrations of Al, As, Cr, and Fe exceeded permissible limits set by the WHO and FAO. In Iran, permissible levels for metals like Cd and Pb are established by national standards; however, for other metals, only international standards are available due to the absence of local regulations (see Fig 2). Our findings indicate that levels of Al, As, Cr, and Fe surpassed these international thresholds. Additionally, Cd concentrations exceeded the national standards in all sampling stations, raising public health concerns.

### Human health risk assessment

**Chronic daily intake.** The analysis for both children and adults revealed that Fe, Zn, and Al had the highest daily intake levels, while Co and V had the lowest (S5 Table). Notably, arsenic was the only element with a daily intake exceeding the permissible limit.

**Non-carcinogenic risk.** Hazard quotient (HQ) and hazard index (HI) were calculated for both age groups. Results showed that HQ values for all metals, except As and Co, were below 1.0, indicating no significant non-carcinogenic risk individually. However, the combined HI exceeded the threshold of 1.0 for both adults and children across all regions, suggesting potential health concerns. The mean HI for children was approximately 1.14 to 1.25 times higher than that for adults (Table 2).

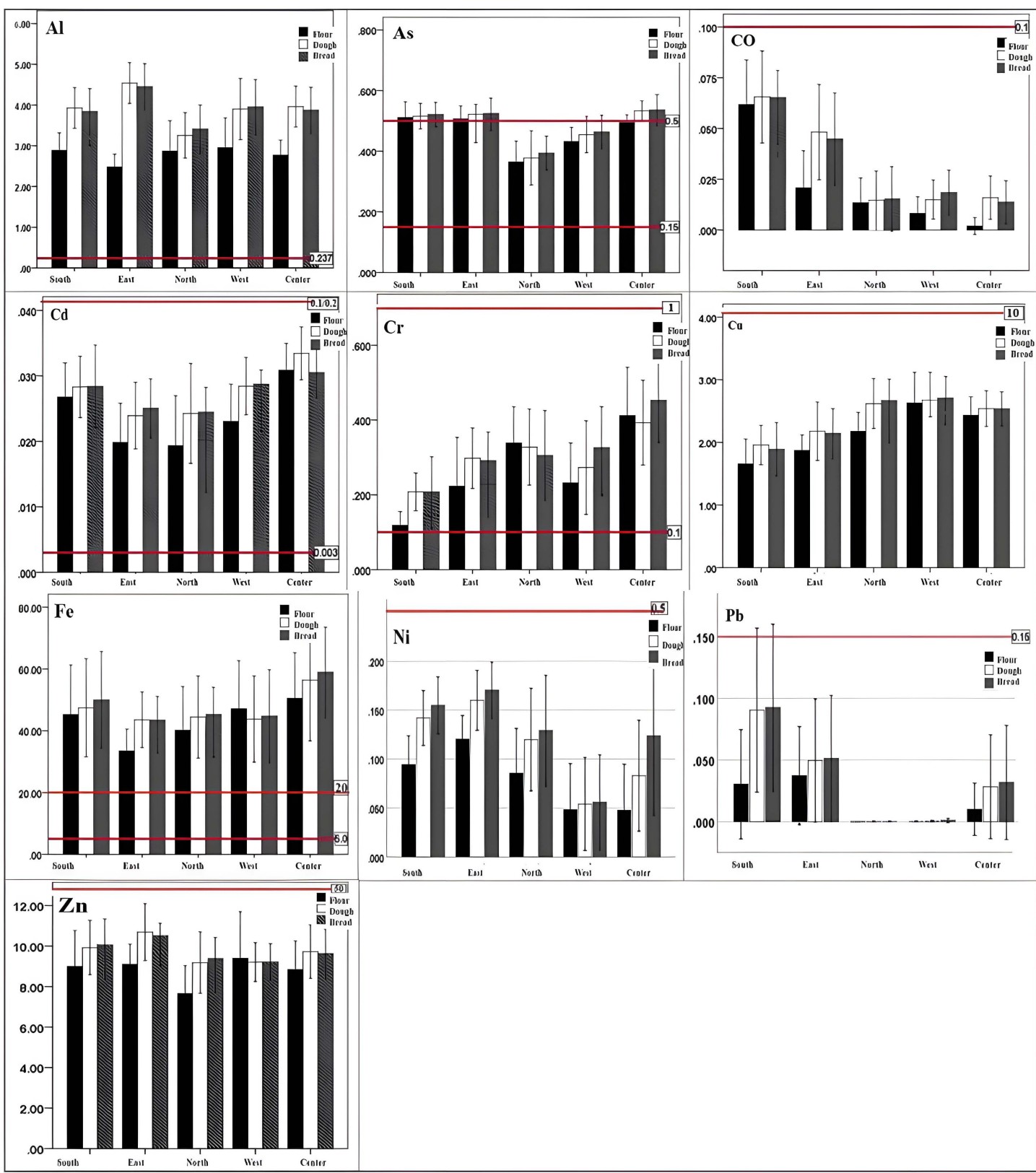

**Fig 2. Comparison of heavy metals concentrations in bread with WHO/FAO standards, with Al, As, Cr, and Fe exceeding limits.**

**Table 2. HI and HQ of heavy Metals in children and adults through bread consumption in all studied areas in Mashhad city.**

| Metals | South | | East | | North | | West | | Center | |
|---|---|---|---|---|---|---|---|---|---|---|
| | Adult | Child | Adult | Child | Adult | Child | Adult | Child | Adult | Child |
| Al | 0.029 | 0.036 | 0.037 | 0.045 | 0.029 | 0.036 | 0.034 | 0.041 | 0.033 | 0.041 |
| As | 5.21 | 6.5135 | 5.066 | 6.33 | 3.955 | 4.943 | 4.633 | 5.792 | 5.361 | 6.702 |
| Cd | 0.170 | 0.213 | 0.150 | 0.188 | 0.121 | 0.152 | 0.157 | 0.197 | 0.183 | 0.229 |
| Co | 1.297 | 0.833 | 0.671 | 0.839 | 0.217 | 0.271 | 0.276 | 0.345 | 0.205 | 0.256 |
| Cr | 0.0006 | 0.0008 | 0.001 | 0.001 | 0.001 | 0.001 | 0.001 | 0.002 | 0.002 | 0.002 |
| Cu | 0.283 | 0.354 | 0.300 | 0.375 | 0.331 | 0.414 | 0.370 | 0.463 | 0.380 | 0.476 |
| Fe | 0.428 | 0.535 | 0.360 | 0.450 | 0.367 | 0.458 | 0.383 | 0.479 | 0.505 | 0.613 |
| Ni | 0.029 | 0.036 | 0.043 | 0.053 | 0.032 | 0.040 | 0.019 | 0.024 | 0.059 | 0.074 |
| Zn | 0.192 | 0.239 | 0.201 | 0.252 | 0.171 | 0.213 | 0.184 | 0.230 | 0.192 | 0.240 |
| V | 0.006 | 0.007 | 0.025 | 0.032 | 0.004 | 0.005 | 0.087 | 0.108 | 0.047 | 0.058 |
| [a]p-Value | <0.001 | <0.001 | <0.001 | <0.001 | <0.001 | <0.001 | <0.001 | <0.001 | <0.001 | <0.001 |
| HI n=90 | 7.6446 | 8.7673 | 6.854 | 8.565 | 5.228 | 6.533 | 6.144 | 7.681 | 6.967 | 8.691 |

a: Statistically significant differences in the concentration of heavy metals in bread samples from various areas were analyzed using the Kruskal-Wallis test.

n: number of samples

**Carcinogenic risk.** The calculated cancer risk (CR) values from bread consumption ranged between $3.61 \times 10^{-3}$ and $4.90 \times 10^{-3}$ for adults, and between $4.50 \times 10^{-3}$ and $6.11 \times 10^{-3}$ for children across all regions. The average CR was approximately $4.42 \times 10^{-3}$ for children and $5.52 \times 10^{-3}$ for adults (Table 3). These values exceed the acceptable risk level of $10^{-4}$, indicating a potential carcinogenic concern.

## Uncertainty and sensitivity analysis

Using Crystal Ball software with 10,000 iterations, the 95th percentile values for HI ranged from 1.50 to 8.94 for children and 1.20 to 7.14 for adults, both exceeding the EPA's permissible limit (HI > 1). The CR for children ranged from $1.21 \times 10^{-3}$ to $3.10 \times 10^{-3}$, and for adults from $9.67 \times 10^{-4}$ to $2.48 \times 10^{-3}$, both surpassing the EPA's acceptable limit of $1 \times 10^{-6}$ (see Fig 3).

**Table 3. Carcinogenic risk of heavy metals in bread in Mashhad for the child and adult populations in various areas.**

| Sites (n=90) | Age category | As | Cd | Total CR (TCR) |
|---|---|---|---|---|
| South | Adult | $4.69 \times 10^{-3} \pm 7.24 \times 10^{-4}$ | $6.47 \times 10^{-5} \pm 2.90 \times 10^{-5}$ | $4.76 \times 10^{-3} \pm 7.53 \times 10^{-4}$ |
| | Child | $5.86 \times 10^{-3} \pm 9.04 \times 10^{-4}$ | $7.46 \times 10^{-5} \pm 3.35 \times 10^{-5}$ | $5.94 \times 10^{-3} \pm 9.38 \times 10^{-4}$ |
| East | Adult | $4.56 \times 10^{-3} \pm 6.92 \times 10^{-5}$ | $6.47 \times 10^{-5} \pm 2.06 \times 10^{-5}$ | $4.62 \times 10^{-3} \pm 8.98 \times 10^{-5}$ |
| | Child | $5.70 \times 10^{-3} \pm 8.65 \times 10^{-4}$ | $6.85 \times 10^{-5} \pm 2.37 \times 10^{-5}$ | $5.77 \times 10^{-3} \pm 8.98 \times 10^{-4}$ |
| North | Adult | $3.56 \times 10^{-3} \pm 1.63 \times 10^{-3}$ | $4.61 \times 10^{-5} \pm 3.68 \times 10^{-5}$ | $3.61 \times 10^{-3} \pm 1.67 \times 10^{-3}$ |
| | Child | $4.45 \times 10^{-3} \pm 2.03 \times 10^{-3}$ | $5.31 \times 10^{-5} \pm 4.23 \times 10^{-5}$ | $4.50 \times 10^{-3} \pm 2.07 \times 10^{-3}$ |
| West | Adult | $4.71 \times 10^{-3} \pm 9.96 \times 10^{-4}$ | $5.98 \times 10^{-5} \pm 2.15 \times 10^{-5}$ | $4.23 \times 10^{-3} \pm 1.02 \times 10^{-3}$ |
| | Child | $5.21 \times 10^{-3} \pm 1.25 \times 10^{-3}$ | $6.88 \times 10^{-5} \pm 2.48 \times 10^{-5}$ | $5.28 \times 10^{-3} \pm 1.27 \times 10^{-3}$ |
| Center | Adult | $4.83 \times 10^{-3} \pm 9.10 \times 10^{-4}$ | $6.95 \times 10^{-5} \pm 1.72 \times 10^{-5}$ | $4.90 \times 10^{-3} \pm 9.27 \times 10^{-4}$ |
| | Child | $6.03 \times 10^{-3} \pm 2.90 \times 10^{-5}$ | $8.01 \times 10^{-5} \pm 1.99 \times 10^{-5}$ | $6.11 \times 10^{-3} \pm 4.89 \times 10^{-5}$ |
| Total | Adult | $4.36 \times 10^{-3} \pm 1.12 \times 10^{-3}$ | $5.95 \times 10^{-5} \pm 2.66 \times 10^{-5}$ | $4.42 \times 10^{-3} \pm 1.15 \times 10^{-3}$ |
| | Child | $5.45 \times 10^{-3} \pm 1.40 \times 10^{-3}$ | $6.85 \times 10^{-5} \pm 3.70 \times 10^{-5}$ | $5.52 \times 10^{-3} \pm 1.44 \times 10^{-3}$ |

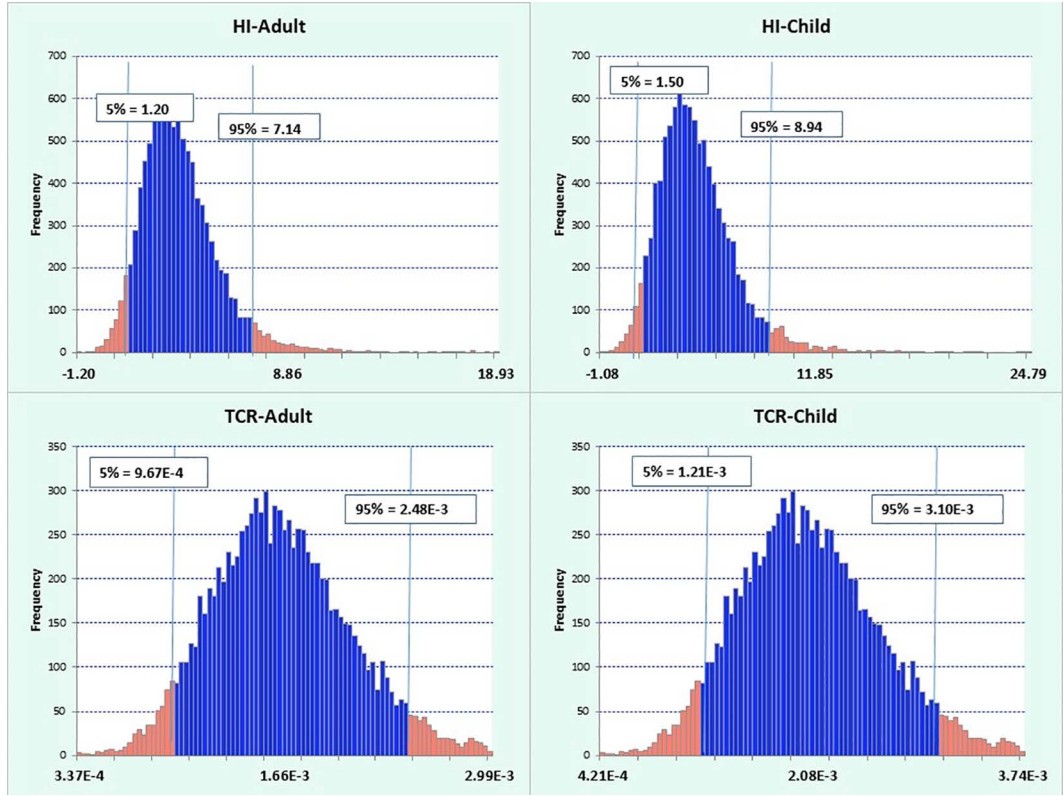

**Fig 3. The cumulative distribution of non-carcinogenic and carcinogenic risk index of heavy metals in children and adults due to bread consumption.**

A sensitivity analysis was performed to evaluate how factors like heavy metal levels, body weight, bread consumption rates, and exposure duration affect the HI and CR. The sensitivity analysis determines how much each variable influences the outcomes. As illustrated in Fig 4, As concentration significantly drove the HI, contributing 69.0%–69.3% to the variance for both children and adults. Other HMs contributed less: Co (10.2%–10.3%), Cr (9.1%–9.2%), Fe (2.5%–2.6%), Cd (1.2%–1.3%), Ni (1.1%), and V, Zn, Al (each <1%). For CR, As accounted for 84.5%–84.6% of the variance in both age groups, with other factors contributing <10%.

These findings highlight that arsenic in bread is the dominant factor influencing both carcinogenic and non-carcinogenic health risks. In contrast, variables such as per capita bread consumption and body weight had relatively minor impacts. Similar conclusions have been reported by Wang et al. [45,46], Sharafi et al. [47], and Liu et al. [48], emphasizing the critical need to control arsenic contamination in cereals to mitigate health risks.

## Discussion

### Heavy metal concentrations

Among the 12 heavy metals analyzed, concentrations of Fe, Al, As, and Cr exceeded standard limits. In contrast, levels of Cu, Ni, Zn, Cd, Co, and V remained within permissible thresholds. Pb and Hg were below the detection limits of the analytical equipment used.

The elevated average concentrations of Fe and Cr beyond standard thresholds may result from soil contamination due to geological sources, overuse of chemical fertilizers on cereal crops, iron fortification of wheat flour, deterioration of

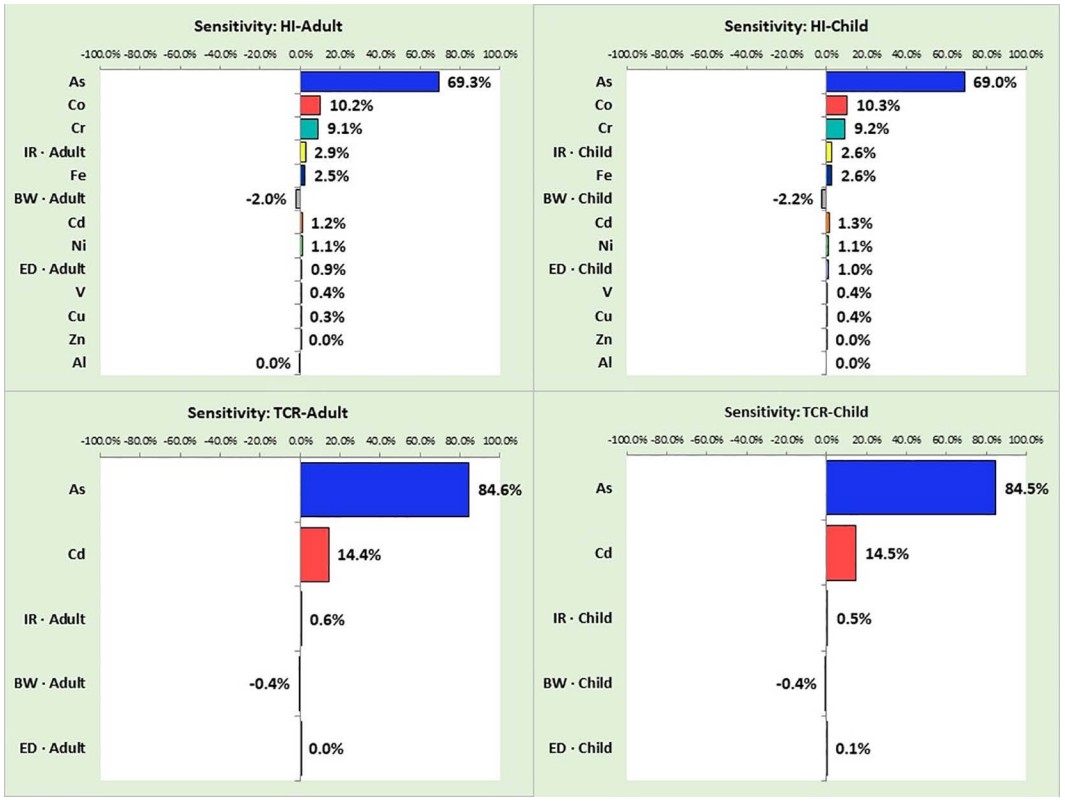

**Fig 4. The impact of various variables on non-carcinogenic and carcinogenic risk indices associated with bread consumption in children and adults.**

processing equipment, and baking processes [49–51]. Naturally, aluminum occurs in environments through processes such as soil erosion, mineral weathering, and volcanic activity. Additionally, anthropogenic activities, including pesticide application, mining, and metallurgical industries, contribute to its presence. Contaminated water, flour, salt, yeast, food additives used in bread, aluminum utensils, and combustion from flame torches can also elevate aluminum levels in bread products [52,53]. Arsenic contamination in bread can originate from pesticide use, organic fertilizers, and irrigation water contaminated with industrial wastewater employed in wheat cultivation. Moreover, metals such as zinc and iron often contain arsenic impurities. Consequently, the use of low-purity alloys in manufacturing equipment and metal tools in contact with dough and bread may serve as sources of arsenic contamination. High-temperature processes involved in bread production facilitate the migration of ions from metal surfaces to the food, while equipment corrosion further increases metal leaching, contributing to contamination [54,55].

A study conducted in Tehran in 2014 reported that Cu, Pb, and Cr concentrations in bread samples were within standard limits. However, iron levels often exceeded WHO guidelines [56], aligning with the current study's findings. Notably, this study also observed slightly higher Pb levels than permissible standards, contrasting with the present results. Investigations in Golestan Province in 2019 [57] and Hamadan City in 2017 [15] revealed that only Cd levels exceeded WHO standards, while other metal concentrations remained within acceptable limits.

Previous research by Khaniki et al. (2004) in Tehran indicated that Cd and Pb levels in bread samples exceeded standard thresholds, whereas Ni concentrations were within permissible limits [58]. Similarly, Feizi et al. (2016) in Chaharmahal and Bakhtiari reported that As, Fe, Cu, Ni, and Cd levels were within safe limits, but Pb concentrations exceeded

permissible thresholds [10]. Haji Mohammad et al. (2013) in Yazd found Pb and Cd levels below maximum allowable limits [59]. Ghoreishy et al. (2018) in Isfahan observed that while Cd levels remained within standards, some samples showed Pb levels above permissible limits [60].

Based on multiple studies across Iran from 2014 to 2019, including cities such as Tehran, Golestan, Hamadan, Yazd, Isfahan, and Kermanshah, most reported Pb and Cd concentrations exceeding safety thresholds, which contrasts with the current study's findings. Variations in heavy metal levels across regions can be attributed to factors such as differences in wheat cultivation practices, processing methods, storage conditions, equipment quality, baking techniques, additive use, and water sources.

Comparative studies from other countries also provide valuable context. For instance, Christopher (2007) in Nigeria analyzed heavy metals in wheat flour and found that manganese, lead, arsenic, and mercury were not detected in all samples, with overall concentrations below acceptable limits [61]. A 2013 study in Ghana evaluated metals like Co, Cd, Fe, Mn, Cu, Zn, Pb, Ni, Cr, As, and Hg, observing significant differences in Fe, Zn, Mn, Cd, and Pb levels, while Co, Ni, Cu, Cr, As, and Hg were below detection limits [51]. The European Food Safety Authority (2012) reported that Cd levels in bread remained below regulatory standards [62]. In Poland, Filon et al. (2013) found that Pb and Cd intake levels, although substantial, stayed within WHO and FAO safety standards [63]. Similarly, a 2014 Nigerian study by Oyekunle et al. measured Co, Cu, Mn, Pb, and Zn, noting that Co, Mn, and Pb exceeded WHO and FAO guidelines, while Cu and Zn did not [64]. In China, Lei (2015) assessed heavy metals in wheat flour, finding most concentrations below Chinese safety thresholds, though approximately 15% of samples exceeded standards [65]. Magomya (2013) in Nigeria reported that all heavy metals measured in bread samples were within permissible limits, except for Fe in two samples [66]. Studies from Ethiopia [67] and Spain [68] similarly reported metal levels within regulatory limits.

Considering these findings, the presence of heavy metals in bread consumed in Mashhad may pose potential health risks. Based on these findings, the presence of heavy metals in bread consumed in Mashhad may pose potential health risks. To mitigate this contamination, the following measures are recommended:

1. Avoid using sewage sludge as fertilizer and prevent the use of urban wastewater for irrigation.

2. Prevent the establishment of industries near agricultural lands.

3. Reduce the excessive use of chemical fertilizers, especially phosphate fertilizers.

4. Implement controls to minimize metal contaminants during wheat processing and flour production.

5. Use indirect flame baking ovens for bread production.

6. Minimize the use of metal compounds, gray cast iron alloys, and steel in conveyor belts of baking ovens, and replace them with polymer or ceramic materials.

### Human health risk assessment

The hazard index (HI) for non-carcinogenic risks associated with dietary intake of heavy metals through bread consumption exceeded 1 for all age groups across all regions of Mashhad, indicating a potential risk of non-carcinogenic effects (Table 2). The mean HI was significantly higher in children (8.047; range: 6.533–8.7673) than in adults (6.567; range: 5.228–7.6446), with statistically significant differences across regions ($p < 0.001$). This difference is primarily due to children's lower body weight and higher relative bread consumption, leading to greater exposure per unit of body weight [69]. Elevated arsenic (As) levels were the primary contributor to the HI exceeding 1 for all age groups, whereas the hazard quotients (HQs) for aluminum (Al), iron (Fe), zinc (Zn), chromium (Cr), copper (Cu), cobalt (Co), nickel (Ni), cadmium (Cd), and vanadium (V) were less than 1, indicating no significant non-carcinogenic risk from these metals.

The carcinogenic risk (CR) associated with As exposure through bread consumption exceeded the U.S. Environmental Protection Agency's (EPA) acceptable risk threshold ($1 \times 10^{-6}$ to $1 \times 10^{-4}$) for both adults and children across all regions, suggesting a high cancer risk (Table 3). The CR results indicate that the incidence of cancer linked to As in bread exceeds one case per thousand individuals, posing a significant public health concern.

To contextualize these findings, we compared our results with prior studies on heavy metal risks in wheat and bread. Ghanati et al. (2019) [70] analyzed grain products in Iran and reported an HI > 1 for As in urban and rural areas, consistent with our findings of elevated As-driven risks in Mashhad. Similarly, Amirabadi et al. (2020) [57] found unsafe levels of Pb and Cd in wheat and flour from Golestan and Mazandaran, though our study did not detect Pb above the limit of detection, highlighting regional differences in contamination profiles. Kianpour and Ardakani (2017) [15] reported an HI > 1 for Cd in wheat for children in Hamadan, whereas Cd posed no significant risk in our study, possibly due to differences in agricultural practices. Internationally, Huang et al. (2008) [3] in China and Bermudez et al. (2011) [8] in Argentina reported HI > 1 for certain metals in wheat, supporting the global relevance of heavy metal risks in staple crops. However, Zheng (2015) [19] and Lei (2015) [65] found no significant non-carcinogenic risks (THQ < 1) for metals in Chinese wheat, suggesting variability in contamination levels and exposure scenarios.

For carcinogenic risks, our findings align with studies by Noori et al. (2020) [71] in Khuzestan, Khodayi et al. (2020) [72] in Isfahan, and Aelmu et al. (2020) [73] in Ethiopia, which reported CR values exceeding EPA thresholds for As in wheat or bread. In contrast, Udowelle et al. (2017) [74] found no cancer risk from Cd and Pb in Nigerian bread, likely due to lower contamination levels. These comparisons highlight that As is a consistent driver of health risks across regions, as observed in our study.

Our results underscore that bread, primarily through flour contamination (83% of heavy metal contribution, S1 Fig), is a significant source of heavy metal exposure in Mashhad, with As posing both non-carcinogenic and carcinogenic risks. These findings emphasize the need for targeted interventions, such as enhanced monitoring of flour quality and agricultural inputs, to mitigate heavy metal exposure through bread consumption [20].

To mitigate heavy metal contamination, interventions targeting flour, baking, and processing are feasible. One promising strategy to combat heavy metal contamination in flour and bread is fortification. By incorporating essential nutrients that can chelate heavy metals, such as calcium or iron, it may be possible to reduce the bioavailability of these toxic elements during digestion [75].

Moreover, alternative baking methods could also play a crucial role in addressing contamination levels. For instance, using baking techniques that involve lower temperatures or shorter cooking times could potentially limit the migration of heavy metals from baking equipment into the final product [76]. Additionally, changing the type of oven used for baking, such as transitioning to electric ovens from traditional wood-fired ovens, could help minimize metal exposure, given the high temperatures and materials typically associated with traditional baking methods [31].

Improved flour processing techniques may also reduce the presence of heavy metals in flour. Techniques such as washing and refining grains before milling can effectively remove some surface contaminants. Furthermore, sourcing flour from regions with lower soil contamination levels and implementing stringent quality control measures in flour production could significantly decrease heavy metal levels in the final products [77].

## Conclusion

The results show that Fe, Al, As, and Cr concentrations in Mashhad's bread exceed national and international standards, posing significant health risks to consumers, primarily due to wheat flour contamination. Non-carcinogenic risk assessment reveals arsenic levels exceeding safe thresholds, with the hazard index (HI) for As indicating a substantial threat. Cancer risk assessment confirms a high risk, with incidence rates exceeding one case per thousand individuals for both children and adults across all Mashhad regions. Given potential unquantifiable limitations, actual risks may surpass these estimates, and with cancer risk ranging from $10^{-3}$ to $10^{-4}$, rising pollutant levels could amplify this threat over time. These findings

underscore the urgent need for policy interventions and necessary measures. It is recommended that local health authorities implement routine monitoring of heavy metal levels in wheat and its processed products to ensure food safety. Additionally, establishing stricter limits on heavy metal concentrations in food products, along with public awareness campaigns about the health risks associated with contaminated bread, could significantly reduce exposure. Furthermore, comprehensive studies should be conducted to identify other potential sources of heavy metal contamination in flour and bread.

## Supporting information

**S1 Fig. The contributions of pathways for the entry of heavy metals into bread.**
(TIF)

**S1 Table. LOD, wavelength and recovery rate of ICP-OES for different elements.**
(DOCX)

**S2 Table. Mean concentrations of arsenic and heavy metals (mg kg−1) in flour, dough, and bread samples collected in this study.**
(DOCX)

**S3 Table. Mean Concentration of Heavy Metals in Various Types of Bread Offered in Mashhad City.**
(DOCX)

**S4 Table. Permissible limits (mg/kg) for heavy metals in food.**
(DOCX)

**S5 Table. Average chronic daily intake of heavy metals through bread consumption for children and adults (mg per kilogram of body weight per day).**
(DOCX)

## Author contributions

**Data curation:** Sajjad Ghasemi, Seyedeh Belin Tavakoly Sany.

**Formal analysis:** Seyedeh Belin Tavakoly Sany.

**Investigation:** Fateme Asadi Touranlou, Sajjad Ghasemi.

**Methodology:** Fateme Asadi Touranlou, Sajjad Ghasemi.

**Project administration:** Seyedeh Belin Tavakoly Sany.

**Supervision:** Mohammad Hashemi.

**Validation:** Mohammad Hashemi.

**Visualization:** Mohammad Hashemi.

**Writing – original draft:** Fateme Asadi Touranlou, Sajjad Ghasemi.

**Writing – review & editing:** Fateme Asadi Touranlou, Mahdi Gholian Aval, Mohammad Hashemi, Seyedeh Belin Tavakoly Sany.

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
