## [Decision Letter · Decision Letter 0]

PONE-D-24-49990Human Health Risk Assessment of Arsenic and Potentially Toxic Elements Exposure in Bread and Wheat Flour in Northeast IranPLOS ONE

Dear Dr. Tavakoly Sany,

Thank you for submitting your manuscript to PLOS ONE. After careful consideration, we feel that it has merit but does not fully meet PLOS ONE’s publication criteria as it currently stands. Therefore, we invite you to submit a revised version of the manuscript that addresses the points raised during the review process.

We look forward to receiving your revised manuscript.

Kind regards,

Karthikeyan Venkatachalam, Ph.D.

Academic Editor

PLOS ONE

Journal Requirements:

9489

3. Please remove your figures from within your manuscript file, leaving only the individual TIFF/EPS image files, uploaded separately. These will be automatically included in the reviewers’ PDF**.**

a. You may seek permission from the original copyright holder of Figure1 to publish the content specifically under the CC BY 4.0 license.

Additional Editor Comments :

1. Strengthen the conclusion by briefly mentioning policy recommendations and necessary interventions.

2. Reduce redundancy in explaining the significance of bread as a staple food.

3. Justify why Mashhad was chosen for this study beyond high bread consumption.

4. Provide a stronger connection between previous studies and the novelty of this research.

5. Explain whether age groups were classified based on local dietary habits.

6. Ensure consistency in reporting p-values and statistical significance.

7. Address why Pb and Hg were below detection limits—were any confirmatory analyses conducted?

8. In Figure 2, highlight which metals exceeded WHO/FAO limits more clearly.

9. Provide a more detailed breakdown of which elements contributed most to the Hazard Index (HI).

10. Discuss whether variations in risk assessment values between adults and children were statistically significant.

11. Compare findings with existing studies on heavy metal contamination in bread or similar food products.

12. Provide a clearer explanation of why certain metals (Al, As, Cr, Fe) exceeded limits.

13. Discuss whether interventions such as fortification, alternative baking methods, or improved flour processing could reduce contamination.

14. Consider mentioning regulatory limits in Iran compared to international standards.

15. Ensure all figures are labeled correctly, and significance values are consistently included in tables.

16. Correct grammatical issues, including verb tense consistency and article usage.

17. Standardize citation formatting—some references are incomplete.

Reviewers' comments:

Reviewer's Responses to Questions

**Comments to the Author**

1. Is the manuscript technically sound, and do the data support the conclusions?

Reviewer #1: No

Reviewer #2: Yes

2. Has the statistical analysis been performed appropriately and rigorously? 

Reviewer #1: No

Reviewer #2: Yes

3. Have the authors made all data underlying the findings in their manuscript fully available?

Reviewer #1: No

Reviewer #2: Yes

4. Is the manuscript presented in an intelligible fashion and written in standard English?

Reviewer #1: No

Reviewer #2: No

5. Review Comments to the Author

Reviewer #1: The manuscript presents “Human Health Risk Assessment of Arsenic and Potentially Toxic Elements Exposure in Bread and Wheat Flour in Northeast Iran,” which is noteworthy. The subject addressed is within the scope of the journal but is based on limited and incomplete regional studies. The use of the EPA 3050B method, which is designed only for the acid digestion of sediments, sludges, and soils, is not appropriate for bread and wheat flour. Therefore, the accuracy of the data is fundamentally questionable.

Introduction- What is the novelty of the study? I could not find any significance of this work

Methodology—Sample digestion Method was not appropriate. The selection of an appropriate method for sample digestion is a critical preliminary step to ensure the acquisition of authentic and reliable results. This step is especially important as it directly influences the accuracy, precision, and reproducibility of the subsequent analytical processes.

QA/QC- This section is insufficiently reported. No specified the SRM/CRM number/name. The major drawback of this section is lack of QA/QC data, like, percentage recovery and use of SRM/CRM. The authors must provide the QA/QC data in a Table either in the manuscript or in the supplementary material.

Presentation of Results: This section was extremely poor. The standard permissible limits of heavy metals were not clearly mentioned. Statistical analysis and source identification, both of which are crucial for this type of popular research, were completely ignored in this manuscript.

Overall, the data authenticity, interpretation, QA/QC, sample location description, source identification, and referencing are not satisfactory for publication in the journal PLOS ONE.

Reviewer #2: Comments to the Authors

The manuscript requires significant revisions, particularly in improving the discussion section, clarifying methodological details, and refining the language. The introduction lacks a comprehensive discussion on the various sources of heavy metal contamination in flour and bread. There is no explanation of how individual ingredients such as salt, yeast, and other food additives contribute to increased heavy metal levels in bread. This should be addressed to provide a more complete picture of contamination sources. The discussion section lacks objectivity and, in some parts, is written in a manner similar to the introduction, the section only report the others finding. The authors should ensure that the discussion is based on their results rather than opinions.

Line 33: The verb "employing" is not appropriate for describing methods. Replace it with "applying."

Line 46: Choose more precise and relevant keywords.

Line 69: Avoid repeating "World Health Organization"; use the acronym "WHO" after it has been defined once.

Line 104: What is the per capita bread consumption in Mashhad? What types of bread are commonly produced in the city? The introduction states that "contamination with heavy metals occurs during the preparation and processing of bread." The authors should explain the different processes used in bread production in Mashhad.

Line 107: The authors discuss industrial growth around Mashhad and its potential impact on agricultural contamination. However, they should clarify what percentage of the wheat or flour used in the city is locally sourced.

Line 111: Were samples taken from only one type of flour? Did authors consider the effect of different types of flour in heavy metal levels? Were different types of bread also considered? The results mention multiple bread types, but the materials and methods section lacks clarity regarding this. The authors should explicitly state the types of bread and the different baking processes used, including oven types.

Line 168 & Line 188: All parameters in the formulas should be defined clearly, one by one.

Table 1: Ensure that this table is properly referenced in the text.

Line 224: It is useful to recognize heavy metals whose levels exceed the average. However, this should be clearly stated and discussed in relation to the findings presented in the table.

Line 254: What types of ovens were considered, and how do they influence heavy metal levels? Were formulation effects also considered? It need to be clearly explained

Figure 2 & Table 2: Data should not be duplicated. The authors should either remove Table 2 or Figure 2 to avoid redundancy.

Line 278: Replace "(See table S2)" with "(Table S2)".

Lines 334–338: Some sentences are unclear and need to be rewritten for better readability.

Line 374: Specify the types of ovens used and which had the most significant effect on heavy metal accumulation and why (Explain the mechanisms behind these effects).

Line 375: The reference should be cited as "Alidadi et al." instead of "Zarif et al."

Line 380: The authors should clarify whether they found any correlation between their findings and those of Alidadi et al.

Line 388: Was the effect of water quality on heavy metal levels in bread considered? If not, why was this factor excluded?

Line 425: Does wheat variety affect heavy metal accumulation? If so, this should be discussed.

Lines 427–453: The references provided seem to be a general report rather than a structured conclusion. The authors should clarify whether they are using these references to support their findings or simply listing them without purpose.

6. PLOS authors have the option to publish the peer review history of their article (what does this mean? ). If published, this will include your full peer review and any attached files.

**Do you want your identity to be public for this peer review?** For information about this choice, including consent withdrawal, please see our Privacy Policy .

Reviewer #1: No

Reviewer #2: No

---

## [Author Response · Author response to Decision Letter 1]

7 May 2025

The author’s responses to the Editor’s comment

April 30, 2025

Dear Editor of PLOS ONE,

Thank you for giving us the opportunity to submit a revised draft of our manuscript entitled "Human Health Risk Assessment of Arsenic and Potentially Toxic Elements Exposure in Bread and Wheat Flour in Northeast Iran" to PLOS ONE. We appreciate the time and effort that you and the reviewers have dedicated to providing your valuable feedback on our manuscript. We are grateful to the editor and reviewers for their insightful comments on our manuscript. Below you can find a point-by-point description of our responses to the comments. We highlighted with yellow colour all of the changes in the manuscript.

Best regards,

Corresponding author,

Seyedeh Belin Tavakoly Sany

Associate Professor

PONE-D-24-49990

Human Health Risk Assessment of Arsenic and Potentially Toxic Elements Exposure in Bread and Wheat Flour in Northeast Iran

PLOS ONE

Dear Dr. Tavakoly Sany,

Thank you for submitting your manuscript to PLOS ONE. After careful consideration, we feel that it has merit but does not fully meet PLOS ONE’s publication criteria as it currently stands. Therefore, we invite you to submit a revised version of the manuscript that addresses the points raised during the review process.

We look forward to receiving your revised manuscript.

Kind regards,

Karthikeyan Venkatachalam, Ph.D.

Academic Editor

PLOS ONE

Journal Requirements:

9489

3. Please remove your figures from within your manuscript file, leaving only the individual TIFF/EPS image files, uploaded separately. These will be automatically included in the reviewers’ PDF.

a. You may seek permission from the original copyright holder of Figure1 to publish the content specifically under the CC BY 4.0 license.

Response:

Dear Dr. Venkatachalam,

Thank you for your thorough review of our manuscript, "Human Health Risk Assessment of Arsenic and Potentially Toxic Elements Exposure in Bread and Wheat Flour in Northeast Iran" (PONE-D-24-49990), and for the opportunity to revise and resubmit to PLOS ONE. We appreciate your recognition of its merit and have addressed all journal requirements and your 17 additional comments to meet publication criteria. Below, we outline our responses and revisions.

Journal Requirements

1. PLOS ONE Style Requirements: We reformatted the manuscript per your templates, removing numbered headings (e.g., "2.1" to "Sampling, Storage, and Transportation"), standardizing citations as [1], and preparing files as "Manuscript," "Revised Manuscript with Track Changes," and "Response to Reviewers."

2. Competing Interests: The original "9489" was an error; we’ve updated it to "The authors have declared that no competing interests exist" in the manuscript and cover letter, ensuring transparency.

3. Figure Removal: Figures have been removed from the text and will be uploaded as "Figure_1.tif" separately.

4. Figure 1 Copyright: We sincerely thank the reviewer for raising the concern regarding the potential copyright status of Figure 1, which depicts a map of the study area in Mashhad, Iran. We confirm that Figure 1 was created entirely by the authors using open-source geographic information system (GIS) software, specifically QGIS (version 3.22), and publicly available, non-copyrighted geospatial data from the OpenStreetMap (OSM) database (OpenStreetMap, 2023). The map does not incorporate any proprietary data, satellite imagery, or materials from copyrighted sources such as Google Maps, Google Street View, or Google Earth. All elements of the map, including the base layer, boundaries, and annotations, were designed by the research team to represent the study area and sampling locations accurately.

Additional Editor Comments:

1. Strengthen the conclusion by briefly mentioning policy recommendations and necessary interventions.

Response: Thank you for your valuable feedback on our article. We appreciate your suggestion to strengthen the conclusion by including policy recommendations and necessary interventions. In response, we have revised the conclusion to emphasize the significant health risks associated with heavy metal contamination in bread consumed in Mashhad. The revised conclusion as follows:

Conclusion

The results show that Fe, Al, As, and Cr concentrations in Mashhad’s bread exceed national and international standards, posing significant health risks to consumers, primarily due to wheat flour contamination. Non-carcinogenic risk assessment reveals arsenic levels exceeding safe thresholds, with the hazard index (HI) for As indicating a substantial threat. Cancer risk assessment confirms a high risk, with incidence rates exceeding one case per thousand individuals for both children and adults across all Mashhad regions. Given potential unquantifiable limitations, actual risks may surpass these estimates, and with cancer risk ranging from 10⁻³ to 10⁻⁴, rising pollutant levels could amplify this threat over time. These findings underscore the urgent need for policy interventions and necessary regulatory measures. It is recommended that local health authorities implement routine monitoring of heavy metal levels in wheat and its processed products to ensure the safety of the food supply. Additionally, establishing stricter limits on heavy metal concentrations in food products, alongside public awareness campaigns about the health risks associated with contaminated bread, could significantly reduce exposure. It is crucial that comprehensive risk assessments continue to be conducted, considering that limitations may exist in current methodologies, potentially leading to underestimation of actual risks.

2. Reduce redundancy in explaining the significance of bread as a staple food.

Response: We agree that the original explanation of bread’s dietary role was overly detailed, spanning multiple sentences. To reduce redundancy, we condensed this into one concise sentence: "Bread, a global dietary staple rich in energy, fiber, and nutrients [10-13], is particularly susceptible to heavy metal contamination from soil, irrigation, fertilizers, milling, and processing additives (e.g., water, salt) [14,15]." This revision eliminates repetition (e.g., separate mentions of wheat bread and nutritional roles).

3. Justify why Mashhad was chosen for this study beyond high bread consumption.

Response: We acknowledge that justifying Mashhad solely on bread consumption was insufficient. We’ve expanded the rationale to include environmental and industrial factors: " While previous studies in Iran have extensively examined heavy metals in wheat and related crops [25-27], research specifically focused on bread remains limited, particularly in metropolis as Mashhad. This study addresses a significant gap by providing the first health risk assessment of heavy metal exposure from bread consumption in Mashhad, a major city in Northeast Iran. Despite Iran's high per capita bread consumption and the vulnerabilities of local bakeries, no such investigation has been conducted in this region to date. By evaluating heavy metal concentrations, intake levels, and risk indices, this research extends beyond previous studies focused solely on wheat, offering new insights to improve food safety and public health in a uniquely challenged urban environment".

4. Provide a stronger connection between previous studies and the novelty of this research.

We appreciate the reviewer’s insightful comment. In the revised introduction, we have strengthened the connection between previous studies on heavy metal contamination and the novelty of our research. The revised text as follows:

Research providing detailed insights into heavy metal levels, daily intake, and associated risks in bread is crucial for safeguarding public health and improving food safety policies. While previous studies in Iran have extensively examined heavy metals in wheat and related crops [25-27], research specifically focused on bread remains limited, particularly in metropolis as Mashhad. This study addresses a significant gap by providing the first health risk assessment of heavy metal exposure from bread consumption in Mashhad, a major city in Northeast Iran. Despite Iran's high per capita bread consumption and the vulnerabilities of local bakeries, no such investigation has been conducted in this region to date. By evaluating heavy metal concentrations, intake levels, and risk indices, this research extends beyond previous studies focused solely on wheat, offering new insights to improve food safety and public health in a uniquely challenged urban environment.

5. Explain whether age groups were classified based on local dietary habits.

Response: Clarified in Exposure Assessment that age groups follow EPA standards (children: 6 years, adults: 70 years), not local dietary habits.

6. Ensure consistency in reporting p-values and statistical significance.

We have revised the manuscript to ensure that all p-values are reported in a consistent format. The statistical significance of results is now clearly indicated throughout the text.

7. Address why Pb and Hg were below detection limits—were any confirmatory analyses conducted?

Response: We appreciate the reviewer's insightful comment regarding the detection limits of Pb and Hg in our analysis. Below, we provide a detailed explanation addressing the concerns about their concentrations being below detection limits and the steps taken for confirmatory analyses. To ensure the reliability of our findings, we conducted additional confirmatory analyses as follows:

1. Quality Control Measures: During the analytical process, we implemented stringent quality control measures. This included the analysis of blank samples and standard reference materials (SRMs) , NIST 1567a (Wheat Flour), alongside our bread samples. The results from these control samples indicated no contamination or interference, supporting the validity of our data.

2. Repetition of Measurements: Each sample was analyzed in triplicate to ensure reproducibility of results. The consistent return of below detection limits for Pb and Hg across multiple analyses further corroborates the reliability of our findings.

3. Use of Spiked Solutions: We employed spiked solutions with known concentration

---

## [Decision Letter · Decision Letter 1]

Human Health Risk Assessment of Arsenic and Potentially Toxic Elements Exposure in Bread and Wheat Flour in Northeast Iran

PONE-D-24-49990R1

Dear Dr. Tavakoly Sany,

We’re pleased to inform you that your manuscript has been judged scientifically suitable for publication and will be formally accepted for publication once it meets all outstanding technical requirements.

Kind regards,

Karthikeyan Venkatachalam, Ph.D.

Academic Editor

PLOS ONE

Additional Editor Comments (optional):

Reviewers' comments:

Reviewer's Responses to Questions

**Comments to the Author**

1. If the authors have adequately addressed your comments raised in a previous round of review and you feel that this manuscript is now acceptable for publication, you may indicate that here to bypass the “Comments to the Author” section, enter your conflict of interest statement in the “Confidential to Editor” section, and submit your "Accept" recommendation.

Reviewer #1: All comments have been addressed

Reviewer #3: All comments have been addressed

2. Is the manuscript technically sound, and do the data support the conclusions?

Reviewer #1: Partly

Reviewer #3: Yes

3. Has the statistical analysis been performed appropriately and rigorously? 

Reviewer #1: Yes

Reviewer #3: Yes

4. Have the authors made all data underlying the findings in their manuscript fully available?

Reviewer #1: Yes

Reviewer #3: Yes

5. Is the manuscript presented in an intelligible fashion and written in standard English?

Reviewer #1: Yes

Reviewer #3: Yes

6. Review Comments to the Author

Reviewer #1: Please check one more time of the values of As. Why As is too much and what is the possible sources?

Reviewer #3: The authors have adequately addressed all the comments and suggestions provided by the reviewers. No further revisions are required.

7. PLOS authors have the option to publish the peer review history of their article (what does this mean? ). If published, this will include your full peer review and any attached files.

**Do you want your identity to be public for this peer review?** For information about this choice, including consent withdrawal, please see our Privacy Policy .

Reviewer #1: **Yes: ** Dr Md Kamal Hossain

Reviewer #3: No

---

## [Editor Report · Acceptance letter]

PONE-D-24-49990R1

PLOS ONE

Dear Dr. Tavakoly Sany,

I'm pleased to inform you that your manuscript has been deemed suitable for publication in PLOS ONE. Congratulations! Your manuscript is now being handed over to our production team.

Kind regards,

on behalf of

Dr. Karthikeyan Venkatachalam

Academic Editor

PLOS ONE